# Genome-Wide Analysis in *Drosophila* Reveals the Genetic Basis of Variation in Age-Specific Physical Performance and Response to ACE Inhibition

**DOI:** 10.3390/genes13010143

**Published:** 2022-01-14

**Authors:** Mariann M. Gabrawy, Nick Khosravian, George S. Morcos, Tatiana V. Morozova, Meagan Jezek, Jeremy D. Walston, Wen Huang, Peter M. Abadir, Jeff Leips

**Affiliations:** 1Department of Biological Sciences, University of Maryland, Baltimore County, 1000 Hilltop Circle, Baltimore, MD 21250, USA; mariann.gabrawy@saintpaul.edu (M.M.G.); nkhos1@umbc.edu (N.K.); gmorcos1@umbc.edu (G.S.M.); mjezek1@umbc.edu (M.J.); 2Biology of Healthy Aging Program, Division of Geriatric Medicine and Gerontology, School of Medicine, Johns Hopkins University, Baltimore, MD 21224, USA; jwalston@jhmi.edu (J.D.W.); pabadir1@jhmi.edu (P.M.A.); 3Program in Genetics, Department of Biological Sciences, North Carolina State University, Raleigh, NC 27607, USA; tvmorozo@gmail.com; 4Department of Animal Science, Michigan State University, East Lansing, MI 48824, USA; huangw53@msu.edu

**Keywords:** aging, personalized medicine, frailty

## Abstract

Despite impressive results in restoring physical performance in rodent models, treatment with renin–angiotensin system (RAS) inhibitors, such as Lisinopril, have highly mixed results in humans, likely, in part, due to genetic variation in human populations. To date, the genetic determinants of responses to drugs, such as RAS inhibitors, remain unknown. Given the complexity of the relationship between physical traits and genetic background, genomic studies which predict genotype- and age-specific responses to drug treatments in humans or vertebrate animals are difficult. Here, using 126 genetically distinct lines of *Drosophila melanogaster*, we tested the effects of Lisinopril on age-specific climbing speed and endurance. Our data show that functional response and sensitivity to Lisinopril treatment ranges from significant protection against physical decline to increased weakness depending on genotype and age. Furthermore, genome-wide analyses led to identification of evolutionarily conserved genes in the WNT signaling pathway as being significantly associated with variations in physical performance traits and sensitivity to Lisinopril treatment. Genetic knockdown of genes in the WNT signaling pathway, *Axin*, *frizzled*, *nemo*, and *wingless*, diminished or abolished the effects of Lisinopril treatment on climbing speed traits. Our results implicate these genes as contributors to the genotype- and age-specific effects of Lisinopril treatment and because they have orthologs in humans, they are potential therapeutic targets for improvement of resiliency. Our approach should be widely applicable for identifying genomic variants that predict age- and sex-dependent responses to any type of pharmaceutical treatment.

## 1. Introduction

The capacity of an organism to resist and respond to a challenge, or its resilience, is of considerable importance in geriatrics as aging is associated with an increased vulnerability to physical challenges. Interventions that target the genes that influence age-specific ability to walk, climb, grip, or to perform other physical challenges are limited. Nevertheless, one gene, the angiotensin-converting enzyme (ACE) in humans, is associated with physical performance and, has been referred to as “the physical performance gene” [1,2,3,4]. Human ACE is an essential enzyme that regulates the renin-angiotensin system [5]. ACE inhibitors (ACE*i*), such as Lisinopril, are commonly prescribed for hypertension and have also been associated with improvement in physical performance in the elderly [6]. These beneficial side effects of ACE*i* may result from alterations in body composition, the metabolism of skeletal muscle, and an improved cardiovascular system [7,8,9]. Interestingly, the beneficial effects of ACE*i* on physical performance traits are not always observed [10,11,12,13] and, in some cases, are proposed to be detrimental [14,15].

One factor that may contribute to the variable effects of ACE*i* on physical performance is genetic variation among individuals. This idea is supported by the fact that variation in the ACE gene alone has been associated with different patient outcomes in response to ACE*i* [16,17]. Incomplete understanding of the genes and genetic networks that might give rise to varying responses of physiological traits to drugs, such as ACE*i*, limits our ability to design effective treatment while minimizing risk in the growing area of personalized medicine.

In this study, we used the fruit fly, *D. melanogaster*, as a model system to identify genetic variants that contribute to differences in individual responses to Lisinopril. *Drosophila* an ideal model for this work as *Drosophila* have an ortholog to mammalian ACE, the target of Lisinopril, angiotensin-converting enzyme, *Ance* [18]. ACE has been associated with age-related decline in muscle function [19]. In addition, the mechanism by which Lisinopril binds to fly Ance is like that of human ACE [20]. Further, our previous study demonstrated significant age- and genotype-specific effects of Lisinopril treatment on speed and endurance, two key indicators of frailty in flies [21] and in humans [22]. Here, we test the hypothesis that responses of these traits to treatment with Lisinopril, depend, in part, on the age and genotype of the individual.

We use genome-wide association studies (GWAs) to identify candidate genes that contribute to natural variation in age-dependent climbing speed and endurance in control and Lisinopril-treated flies. In a manner similar to endurance measures in humans, we measured the distance an individual traveled within a defined period as an indicator of endurance. For each GWA, we used 126 genotypes from the *Drosophila* Genetic Reference Panel (DGRP) [23]. Although this panel consists of 205 genotypes, many of these are short-lived and did not produce enough individuals to measure at old age and so were not used in our study. Results from the GWAs were used to identify genetic networks that contributed to the variation in age-specific response to Lisinopril treatment of each trait using the sensitivity index of Falconer [24]. This index measures the magnitude and direction of the response of a trait to treatment relative to the control for each genotype. Our network analyses implicated genes in the WNT signaling pathway as important for regulating age-specific physical performance and sensitivity.

To validate a role for WNT signaling and the influence of ACE*i*, we tested the effects of altered expression of four genes in the WNT signaling pathway, *Axin* (*Axn*), *frizzled* (*fz*), *nemo* (*nmo*), and *wingless* (*wg*) on age-specific physical performance ability in Lisinopril-treated and control flies. Our results support the role of WNT signaling for regulating age-specific climbing speed and its response to ACE*i* treatment. We highlight that our GWAs using the sensitivity index as a phenotype can be applied to any medication of interest and this provides a unique and powerful approach to identifying polymorphisms that will predict the response to drug treatment and so will be useful to advance the field of personalized medicine.

## 2. Methods

### 2.1. Drosophila Stocks, Maintenance, and Drug Treatment

Virgin males of 126 genotypes from the *Drosophila* Genetic Reference Panel [23,25,26] were used for all physical performance assays. We used virgin flies to avoid any potential physiological effects of mating on aging rates. Control groups were fed standard food medium. Treated groups were fed 1 mM Lisinopril (Sandoz Pharmaceuticals; Princeton, NJ, USA), which was homogenously mixed into the fly food. The 1 mM dose was determined as the optimal dose using a dose-response curve in a previous study [21]. Lisinopril treated flies were maintained on food containing Lisinopril from the time of eclosion until the day they were tested.

Flies were maintained in vials at 25 °C and approximately 55% relative humidity under a 12-h light and dark cycle. All physical performance assays were completed between 8:00 a.m. and 2:00 p.m.

### 2.2. Physical Performance Assays

Climbing speed and endurance assays were used to test physical performance in young and old flies as previously described [21]. In brief, an individual fly was aspirated into the bottom of a Costar^®^ 25-mL IN 2/10 serological pipet which was marked at nine and 27 cm. We measured the time it took, in seconds, for an individual to reach the nine-centimeter mark (climbing speed) and measured the distance, between the nine cm mark and 27 cm mark that an individual traveled in 15 s (endurance). The “distance travelled” in 15 s was chosen instead of time because many of the older flies do not reach the 27 cm mark due to stopping or losing grip [21]. Because the DGRP lines exhibit significant differences in body size [27] flies were individually weighed after each assay for use as a covariate in statistical analysis. We did this to ensure that variation among the lines in body size was not confounded with variation in physical performance. Each fly was tested in only one measure of performance at a given age and drug treatment, so no measurement was repeated on an individual fly. We measured the climbing speed and endurance (distance travelled) of 30 flies per genotype, age, and treatment combination. Physical performance traits of the 126 DGRP genotypes were analyzed in ANCOVA (PROC GLM, SAS V9.3) using fly mass as a covariate in the following model: y = c + m + g + t + a + all interactions + b(a) + *ε*, where y is the phenotype, c is a constant, g tested for differences among DGRP lines, t tested the effects of Lisinopril treatment, a is the effect of age, b is the block effect nested within each age, and *ε* is error. None of the interactions between mass and the main effects were significant, so interaction terms involving mass were dropped from the model. We calculated genetic correlations between control and Lisinopril treated flies within each age (week 1: *r_GT1,_* and week 5: *r_GT5_*) and within each treatment across ages (control: *r_GAC_*, Lisinopril: *r_GAL_*). The genetic correlation was calculated as *cov_12_/σ_L1_σ_L2_* where *cov_12_* is the covariance among line means between treatments or ages and *σ_L1_* and *σ_L2_* are the square roots of the variance components across lines from the reduced model analyses by treatment and age [28].

### 2.3. Genome-Wide Association Tests of Physical Performance and Sensitivity to Lisinopril

To identify candidate SNPs that contribute to variation in the climbing speed, endurance distance, and the sensitivity of each trait to drug treatment we carried out GWA analyses using the DGRP2 web tool (http://dgrp2.gnets.ncsu.edu/, accessed on 20 January 2020). With respect to climbing speed and endurance, we first calculated the least squared line means of each trait in each age and drug treatment combination using ANCOVA to adjust for body mass. We then submitted the least squared means of each line at each age and treatment to the DGRP analysis pipeline (http://dgrp.gnets.ncsu.edu/, accessed on 20 January 2020). Additionally, we calculated the sensitivity index of Falconer [24] for each trait at each age. This method calculates a sensitivity value for each genotype by taking the difference in the average trait value (climbing speed or endurance) between Lisinopril treated and control flies of each genotype and dividing this value by the average difference in treated and control flies across all genotypes. This index provides information on both the magnitude and direction of the response of each trait to Lisinopril. Positive sensitivity values indicate that climbing speed or endurance is enhanced by Lisinopril; negative values indicate that Lisinopril has a negative effect on the trait. We used the DGRP2 web tool to identify polymorphisms associated with the sensitivity of climbing speed and endurance to Lisinopril treatment. In these GWAs, we used the option to include two phenotypes in the analysis, one of which was the sensitivity phenotype and the other the actual trait value of each line in control or Lisinopril treatments. By using the trait value as a cofactor in the sensitivity analyses the resulting polymorphisms associated with sensitivity are those that are significant after accounting for the effects of each SNP on the trait value in each treatment. GWA was completed on 126 genotypes based on the sequence data available at the time of each analysis. The DGRP Freeze 2 GWA analysis uses linear model ANOVAs on 1,887,374 SNPs using the model y = u + m+ ε, where y is the phenotype, m is the effect of the SNP and ε is the error variance [26]. Candidate SNPs associated with each respective phenotype were based on a nominal cutoff of *p* < 10^−5^.

### 2.4. Network Analyses of Physical Performance and Sensitivity to Lisinopril

To prioritize candidate genes for follow-up study, we performed network analyses in the igraph package in R (R Core Team) [29] using genes from each of the GWA analyses. For network analysis within each age, we combined the genes identified as candidates from GWA on flies from both control and Lisinopril treatments. We identified computationally predicted networks of genetically interacting genes, allowing one missing gene in between the candidate genes (i.e., one gene connecting two candidate genes, but was not one carrying a variant associated with the trait). We were more stringent in our use of candidate genes for network construction than for the GWA and used genes significant at *p* < 10^−6^ in the GWA. We mapped candidates to physical and genetic interaction databases downloaded from Flybase release r5.57. Genes in these networks are represented as nodes, whereas edges between nodes represent interactions. We extracted subnetworks from the global networks whose edges either directly connected candidate genes or were bridged by one gene that was not in the list of candidate genes. We tested whether the largest subnetwork was significantly greater than would be expected by chance using a permutation procedure [30]. Briefly, we randomly selected *n* genes that could be mapped to the global networks, where *n* is the number of significant genes mapped to the global network. The size of the largest subnetwork was then computed. This procedure was repeated 1000 times, and the *p*-value was calculated as (A + 1)/1001, where A was the number of permutations in which the size of the largest subnetwork was equal or greater than the size of the largest subnetwork with the observed gene list. Human orthologs were obtained using the DRSC Integrative Ortholog Prediction Tool with all available prediction tools, excluding low scores of less than 2 (DIOPT, version 5.4; http://www.flyrnai.org/diopt, accessed on 20 May 2020). A gene interaction network for human orthologs was constructed using R-Spider (http://www.bioprofiling.de, accessed on 20 May 2020).

### 2.5. Validation of Genes Associated with Physical Performance and Sensitivity to Lisinopril Using Muscle-Specific RNAi

We independently tested the effect of candidate genes on age-specific climbing speed and endurance using the GAL4-UAS system in *Drosophila* to activate RNAi against each gene (Table 1). We altered expression of candidates in muscle tissue using the muscle-specific driver *dj667* [31]. The knockdown efficiency varied among the genes, ranging from 0.25 fold (or 75% decrease for frizzled, wingless) to 0.79 fold (21% decrease for nemo) (Appendix A. Genes were chosen based on their effect size, *p*-value, human orthologs, availability of RNAi stocks, and network analyses. Three of the genes, *fz*, *nmo,* and *wg*, were “non-hub” genes in the network affecting climbing speed at old age and in the network affecting sensitivity at old age. In both cases, “non-hub” indicates that there were ten or fewer connections to other genes in the network. One gene, *Axn*, was identified as a “hub” candidate gene and was connected to several other genes in these networks. All four genes are in the WNT signaling pathway which is known to have roles in muscle stem cell development and maintenance and aging [32,33].

As a test of the age-specific effects identified in the GWA, we compared the effect of the knockdown of each gene on both traits at one and old age. We also compared age-specific climbing speed and endurance of RNAi knockdown flies on control food and Lisinopril-treated food to test the hypothesis that the effects of Lisinopril are influenced by the expression of these candidates.

To generate the experimental lines, we crossed males of the muscle-specific driver with females of each of eight RNAi lines, two different RNAi stocks for each gene, and the control line *attP2* (*dj667*-GAL4 × UAS-RNAi; *dj667*-GAL4 × UAS-*attP2* control line, *n* = 20 per line). RNAi lines were generated by the Transgenic RNAi Project (TRiP) at Harvard Medical School (http://www.flyrnai.org, accessed on 20 June 2020) (Table 1). We used two stocks per RNAi construct to control for potential off target effects and effects of transgene insertion site on the phenotype. We used the standard genetic background control line for these stocks that also contains the *attP2* (stock #36303) landing site, as designated by the TRiP project.

We used quantitative real-time reverse transcriptase polymerase chain reaction (qRT-PCR) to verify the knockdown of the targeted mRNAs using RNAi. Male fly offspring of crosses between each RNAi or control line and the GAL4 driver line were flash frozen with liquid nitrogen at one week of age and stored at −80 °C prior to RNA extraction. RNA was extracted from homogenized tissue of 10 males per strain using the RNeasy Mini Kit from Qiagen. DNA was removed from the samples using the TURBO DNA-free™ Kit (ThermoFisher Scientific, Waltham, MA, USA). cDNA was synthesized using a BioRad iScript™ cDNA Synthesis Kit. 1X iTaq™ Universal SYBR^®^ Green Supermix (BioRad, Hercules, CA, USA) was mixed with 1 µL of the newly synthesized cDNA and 0.5 µM of the appropriate forward and reverse primers. Real-time amplification was performed on a Biorad CFX384 Real-Time Detection System. Three biological replicates were run for every reaction, each with three technical replicates. Relative expression values were first normalized to *Ribosomal Protein L32* (*rp49*) expression levels. The fold change in expression was then calculated and averaged (Appendix A). Primers for *Axn*, *fz*, *nm*, *wg* and *rp49* were designed according to the fly primer bank (http://www.flyrnai.org/cgi-bin/DRSC_primerbank.pl, accessed on 20 August 2020) (Table 1).

To test the effects of individual candidate genes on each trait in the RNAi experiments, we carried out ANOVA using the model y = g + *ε*, where y is the phenotype, g is the genotype of the cross and *ε* is the error. Each analysis was followed by a post hoc Dunnett’s test which compared offspring of the crosses from each RNAi and control (*attP2*) line.

## 3. Results

### 3.1. Genetic Variation in Response and Sensitivity to Lisinopril Treatment—Climbing Speed

We found a significant effect of age, irrespective of treatment, on climbing speed. Young flies climbed 50% faster than old flies (F_1, 14,435_ = 3126.21, *p* < 0.0001; Week 1: 1.54 + 0.01 cm/s, Week 5: 0.77 + 0.01 cm/s) (Appendix A). The effect of age on climbing speed also depended on genotype (age by genotype interaction: F_125, 14,435_ = 6.22, *p* < 0.0001, Appendix A).

Flies treated with Lisinopril, independent of age and genotype, climbed 8% faster than untreated flies (F_1, 14,435_ = 42.22, *p* < 0.0001; Lisinopril-treated mean: 1.20 + 0.01 cm/s, Control mean 1.10 + 0.01 cm/s). However, the effect of Lisinopril treatment also varied significantly among genotypes (F_125, 14,435_ = 2.02, *p* < 0.0001) and the three-way interaction between age, Lisinopril treatment, and genotype was also significant (F_125, 14,435_ = 1.26, *p* = 0.0200) (Figure 1A,B). This suggests that the effect of Lisinopril treatment on climbing speed depended on age and genotype.

To examine the genotype-specific responses to Lisinopril at each age we used the sensitivity index of Falconer [24], which is defined as the difference between Lisinopril treated and control flies in a quantitative trait (climbing speed in this case) for each genotype divided by mean difference across all genotypes. Based on this index, genotypes exhibited substantial variation in their sensitivity to Lisinopril. While most genotypes climb faster (positive sensitivity index) when treated with Lisinopril several lines climbed more slowly (Figure 1C). Thus, the magnitude and the direction of the response to Lisinopril depended on genotype. Genotypes also exhibited a greater range of sensitivity to Lisinopril treatment at older age (−5.78 to +11.42) than at younger age (−3.96 to +9.07; Figure 1C, Appendix A). Not only was there a greater range of response to Lisinopril treatment at old age, but more genotypes exhibited a large response to Lisinopril at old age than at young age (Appendix A).

### 3.2. Genetic Variation in Response and Sensitivity to Lisinopril Treatment—Endurance

Endurance varied significantly among genotypes (F_125, 14,310_ = 4.54, *p* < 0.0001) and declined with age (F_1, 14,310_ = 2384.07, *p* < 0.0001) (Appendix A). Younger flies climbed over twice the distance that older flies reached during the 15 s interval (young flies: 16.57 ± 0.14 cm, old flies: 7.34 ± 0.12 cm) (Appendix A).

Lisinopril treatment, independent of age and genotype, positively influenced endurance (F_1, 14,310_ = 52.68, *p* < 0.0001) as flies treated with Lisinopril climbed 10% farther in the 15 s interval than controls (Lisinopril-treated mean: 12.53 ± 0.14 cm/s, Control mean 11.39 ± 0.14 cm/s). However, the effect of Lisinopril treatment varied significantly among genotypes (drug treatment by genotype interaction, F_125, 14,310_ = 1.75, *p* < 0.0001). There was also a significant three-way interaction (F_125, 14,310_ = 1.62, *p* < 0.0001), indicating that the effect of age on endurance depended both on genotype and Lisinopril treatment (Figure 2A,B, Appendix A). Using the sensitivity index described previously, we found that genotypes were differentially sensitive to Lisinopril treatment at each age (Figure 2C, Appendix A). As we found for climbing speed, genotypes exhibited a greater range of sensitivity to Lisinopril treatment at older age (−6.32 to +11.99) than at young age (−5.92 to +8.68; Appendix A).

### 3.3. GWA of Effect of Lisinopril Treatment on Physical Performance and Sensitivity

We carried out GWA of the effect of Lisinopril on climbing speed (Appendix A), endurance (Appendix A), and the sensitivity of age-specific speed and endurance to Lisinopril treatment (Appendix A). For GWA of each phenotype, age, and treatment, we identified the number of indels, SNPs, and number of genes associated with the trait (Table 2, Appendix A). By convention, we considered candidate SNPs as those that are within or nearby (less than 5000 bp away from) a gene affecting the trait.

### 3.4. Overlap of Candidate Genes Influencing Physical Performance and Sensitivity

The GWAs for physical performance revealed limited overlap in candidate genes between control and drug treated flies within each age or across ages within each treatment group. While this incomplete overlap undoubtedly reflects the limited statistical power of the DGRP lines, it could also reflect age- and treatment-dependent allelic effects on climbing speed and endurance. For climbing speed, 27 out of 114 total genes were identified as candidates affecting both young control and Lisinopril-treated flies (Appendix A). The genetic correlation for climbing speed between treatments at young age was positive but relatively low, *r_GT1_* = 0.24. This low correlation reflects the change in the rank order of the line means in climbing speed across treatments (Figure 1A,B) and supports the finding of limited overlap in significant polymorphisms between treatments. Comparing the candidate genes at old age for climbing speed, only 14 of 128 genes were found in common across treatments (Appendix A). As was the case at young age, the genetic correlation across treatments at old age was low and positive, *r_GT5_* = 0.22.

For endurance, of 9 candidates identified for young flies, only two were found in common between control and Lisinopril treated flies. Both were unnamed genes *CG31013. CG43955* (Appendix A), and the genetic correlation low and positive, *r_GT1_* = 0.28. At old age, only two candidates out of a total of 85 were common between treatments, *Eip78C*, and *caps* (Appendix A), with the genetic correlation low and positive, *r_GT5_* = 0.26.

Comparing the lists of candidates within treatments and across ages for both traits, the only candidates that were identified in both ages were those for climbing speed in the control condition. In this case, only one candidate gene, *sima*, was associated with climbing speed at both ages (Appendix A). The genetic correlation across ages for climbing speed in the control and Lisinopril treated groups was very low *r_GAC_* = 0.11, and *r_GAL_* = 0.08, respectively. The genetic correlation for endurance across ages was also low and positive for the control *r_GAC_* = 0.22, and for the Lisinopril treated flies *r_GAL_* = 0.20.

The GWA for sensitivity, consistent with the GWA of climbing speed, revealed little to no overlap in candidate polymorphisms influencing the sensitivity of these traits to Lisinopril treated flies with those associated with the traits themselves. For climbing speed in young flies, only one candidate was associated with sensitivity and climbing speed (*CG43733*) which was identified in the control treatment (Appendix A). At old age, two candidates for climbing speed in the control treatment were also identified as sensitivity candidates, *esn, cpo* (Appendix A).

For endurance, no genes identified as candidates affecting sensitivity of young flies were found in common with those influencing endurance. At five weeks of age, one candidate was shared between sensitivity and endurance, *Eip63E* (Appendix A).

### 3.5. Network Analysis of Climbing Speed

Although the age and treatment specific GWAs revealed many non-overlapping genes, they may belong to the same genetic networks. We first analyzed networks of genes associated with variation in climbing speed, separately at both ages, and then across ages. To do this, we combined genes identified by GWA when flies were maintained on control or Lisinopril-containing food and asked whether they were enriched for genes that were known to interact with each other. Analysis of climbing speed candidates from young flies revealed a significant network comprised of 33 interacting genes with 17 candidate genes and 16 computationally recruited genes; *Snr1*, which was identified as a candidate gene in the GWA, is the most interconnected gene in the network (Appendix A).

While gene networks for young age are interesting, of particular focus are the candidates associated with climbing speed at old age for many reasons. First, Lisinopril had smaller effects on physical performance in young flies than in old flies. Second, we noted that within the Lisinopril-treated groups, the number of unique genes to old age was more than double of that for young age (Appendix A). Third, when analyzing the sensitivity group, the sensitivity group has a markedly higher number of genes at old age than at young age (Appendix A).

We identified a network of 28 candidate genes and 60 computationally recruited genes (Appendix A) associated with climbing speed at old age. Ninety percent of these genes have known human orthologs. Four of the candidate genes (*Antp, Axn, numb, tup*) and 13 recruited genes (*arm*, *dpp*, *Egfr*, *fz*, *hh*, *Mmp2*, *mys*, *pnt*, *Ras85D*, *shg*, *sli*, *Ubx*, and *wg*) have been associated with heart development. Of note, several genes in this network are involved in the Wg/WNT signaling pathway including, but not limited to, *Axn*, *fz*, *nmo*, and *wg*.

### 3.6. Network Analysis of Endurance

The analyses of candidate genes associated with variation in endurance revealed a network comprised of nine interacting genes with five candidate genes and four computationally recruited genes for young flies (Appendix A). We identified a network of seven candidate genes and four missing genes associated with variation in endurance in old flies (Appendix A). Among the genes in these networks, 70% have known human orthologs. Next, we created genetically interacting network of candidate genes associated with variation in endurance in both young and old flies. We found a network of 50 interacting genes with 16 candidate genes (Appendix A).

Among the genes in the aforementioned networks, 70–90% of the fly genes have human orthologs. As such, we were able to construct a human genetic interaction network based on the *Drosophila* interaction networks associated with variation in climbing speed and endurance together (Appendix A). Sixteen genes formed the network associated with climbing speed (Appendix A). PAK1, PAK2 and PLXNB1 were the most interconnected genes. The resulting network of orthologous human genes associated with variation in endurance in flies consist of 18 orthologs for corresponding genes from the *Drosophila* network (Appendix A). CDC42 and MAP2K1 are the most interconnected genes in the network.

### 3.7. Network Analysis of Sensitivity to Lisinopril

We identified a network of 19 candidate genes with 31 computationally recruited genes associated with sensitivity at five weeks of age for climbing speed (Appendix A). Ninety-four percent of these genes have known human orthologs. Four of these candidate genes (*Axn*, *fz2*, *prickle*, *Star*) are associated with 22 computationally recruited genes (Appendix A), some of which are involved in heart function and in WNT signaling (*wg*, *nmo*, *arm*, *fz*). This result is consistent that of the network analysis for climbing speed of control and Lisinopril-treated flies at old age (Appendix A). We also identified a network of 15 candidate genes with 24 computationally recruited genes associated with sensitivity of flies to Lisinopril treatment at young age (Appendix A). Ninety-two percent of these genes have known human orthologs. Three of the candidate genes (*Rho1*, *klumpfuss*, *antennapedia*) were associated with 23 computationally recruited genes (Appendix A). Networks for sensitivity at young age are provided in Appendix A.

### 3.8. Muscle-Specific RNAi Implicates Genes in the WNT Signaling Pathway as Mediating the Effects of Lisinopril on Physical Performance

To confirm the roles of the candidate genes in mediating the effects of Lisinopril on physical performance, we knocked down four genes in the WNT signaling pathway in muscles by muscle-specific RNAi. We used two independent RNAi stocks for each gene to ensure that the effects of RNAi knockdown are replicable. In the *attP2* (control) line, flies treated with Lisinopril generally climbed faster at each age (Figure 3A–D) however this effect was only formally significant at old age (*p* = 0.0132) (Figure 3B). In contrast, the climbing speed of untreated RNAi-*gene* lines generally resembled that of each RNAi line that was treated with Lisinopril (Figure 3A–D). This suggests that the positive effect of Lisinopril on climbing speed is mediated by the expression of these genes with the possible exception of *wingless*. In the case of *wingless,* flies in three of the four RNAi experiments with this gene still exhibited a positive (although not significant) response to Lisinopril. We also note that in some cases there is inconsistency between RNAi stocks of the same gene. This may reflect variation in the degree to which the gene was knocked down among lines and/or position effects of the insertion of the RNAi construct itself. Treatment with Lisinopril did not significantly affect endurance of the *attP2* control line although the mean endurance was slightly higher with Lisinopril treatment. The endurance of all 16 untreated RNAi-*gene* lines closely resembles that of Lisinopril-treated flies for all six RNAi stocks at old age (Appendix A).

## 4. Discussion

In this study, we used GWAs with 126 genotypes of *Drosophila* to identify genes and genetic networks influencing age-specific physical performance and the sensitivity of these traits to Lisinopril treatment.

There are three key findings of this study. First, we found many polymorphisms that contributed to the variation in male climbing speed and endurance at each age, in control and Lisinopril treatments as well as polymorphisms that contribute to the variation in the response of each genotype to Lisinopril treatment. Interestingly, many polymorphisms did not overlap between treatments or across ages. While low statistical power undoubtedly contributes to this lack of overlap, the results of the GWAs combined with the low genetic correlations across ages and treatments lends support to the hypothesis that the contributions of individual alleles to each trait are sensitive to the environmental conditions as well as age. These condition- and age-dependent allelic effects on these traits will need to be further validated in independent experiments. Second, we used a new approach to identify polymorphisms that would predict the sensitivity of individual genotypes to drug treatment and found that polymorphisms which predict sensitivity to drug treatment were different from those that affect the traits themselves. Finally, we found that genes in the WNT signaling pathway are major contributors to variation in physical performance.

While there was limited overlap in polymorphisms affecting both traits across treatments or ages, some candidates affecting climbing speed and endurance were identified repeatedly across ages and treatments (Appendix A). Many of these genes have been implicated in locomotion and/or muscle development, maintenance, and muscle function in other studies. For example, the genes *htt* and *iab-8* (also known as *Abd-B*) were associated with variation in climbing speed in both the young control and Lisinopril treatments. *htt* (*huntingtin*), the *Drosophila* ortholog of the huntingtin gene (HTT) in humans, has been used as a model in the study of Huntington’s [34] and Parkinson’s disease [35] and is important for maintaining mobility in adult flies [36]. *Abd-B* has been implicated in muscle development [37]. The genes *Antp* (*Antennapedia*), *numb*, *BicD*, *shakB* and *slowdown* were all associated with variation in climbing speed in old flies in both control and Lisinopril treatment. *AntP* is a member of the Antennapedia *HOX* gene complex and is involved in muscle cell fate specification [38]. *numb*, whose Human ortholog is NUMBL, is an inhibitor of Notch signaling and has also plays a role in cell fate specification during development of muscle and heart as well as that of the nervous system [39]. Disruption of *BicD* produces defects in locomotion [40]. The gene *shakB* produces an innexin protein. Innexins are important in forming gap junctions that allow the passage of ions and small molecules between cells. In adult flies, *shakB* is expressed in tergotrochanteral muscle motor neurons as well. Mutations in this gene cause defects in jump response [41], light response [42], and flight capability [43]. *slowdown* is involved in muscle attachment [44]. It is important to note that because variants in these genes were identified as candidates in independent GWA experiments in our study, they are likely causal polymorphisms affecting these traits and so potential targets for therapeutic treatment.

To our knowledge, no other studies have mapped genes influencing climbing speed nor endurance. Only one study [45] has mapped genes influencing negative geotaxis behavior, a similar phenotype to climbing speed; none of the candidate genes in that study were identified as candidates in our study.

Our second key finding is that there are significant genetically based differences in sensitivity to Lisinopril treatment. Interestingly, genes that were associated with the magnitude and direction of response to drug treatment were not the same as those that were associated with variation in the traits themselves. Of course, the limited power of our experiment prevents us from making a definitive statement about the degree to which genes regulating the response of these traits to the drug overlap with those genes affecting the traits themselves. Future work is needed to test this hypothesis. However, if confirmed, this would have profound implications for personalized medicine. Knowledge of the genotype of individuals at genes that regulate the direction and magnitude of the drug response could be invaluable for producing a positive outcome of the treatment. In fact, this information could be more useful to predict successful treatment than knowledge of the polymorphisms that give rise to individual variation in the trait being treated. Extensions of this approach using other drugs and validating the effect of individual genes on drug sensitivity is a major focus of our future work.

Our third key finding is that genes in the WNT signaling pathway are strongly implicated as contributors to variation in climbing speed and the sensitivity to drug treatment. WNT signaling is important in development and stem cell maintenance and has been associated with age related deterioration of muscle function [32]. The WNT pathway regulates many aspects of the phenotype that could influence physical performance such as the development of the nervous system, neuromuscular junctions, and skeletal muscle development [46,47,48]. Likewise, our genetic network analysis point to the WNT signaling pathway as important for resilience to age-related decline in physical performance. Genes in the WNT signaling pathway, such as *Axn*, *fz*, *fz2*, *wg*, *Rpl35A*, and *nmo*, appeared in networks affecting both climbing speed and endurance, particularly speed at old age. Additionally, *Axn* and *fz* appeared in networks affecting sensitivity to Lisinopril treatment. Further support comes from our previous study, using RNAseq on a subset of the 126 genotypes, in which we show that Lisinopril treatment alters expression of genes in the WNT signaling pathway [21].

Knockdown of *Axn*, *fz*, *nmo*, and *wg* expression in skeletal muscle either eliminated or greatly diminished the positive effects of Lisinopril on climbing speed. In fact, treatment with Lisinopril consistently improved speed only in the control flies that expressed these genes at normal levels. These RNAi experiments support the findings of the GWA that polymorphisms in these genes contribute to the observed differences in climbing speed, and response to Lisinopril treatment (Appendix A).

*Axn*, *fz*, *nmo*, and *wg* are evolutionarily conserved from flies to humans and the human genes share many biological functions with those of flies. For example, both *wg* and WNT genes are involved with regulation of cell death [49] and positive regulation of cell proliferation [50]. Of note, *Axn*, *fz*, and *wg* are involved in heart development and *nmo* is involved in regulation of synaptic growth at neuromuscular junction [51]. Human FZD1 is involved in response to drug treatment [52], NLK is involved in a number of pathways, including Wnt [53] and Notch [54] signaling.

Lastly, we provide support to the limited number of studies in vertebrates which indicate a connection between RAS pathway and WNT pathway. One such study showed that treatment with a different ACE*i*, Irbesartan, improves muscle regeneration from injury by reducing WNT signaling by decreasing expression of a WNT signaling activator, C1q [55]. Although flies do not have any known orthologs of C1q, the finding that decreased signaling by WNT abolished beneficial effects of Lisinopril support this connection. That is, ACE*i* reduces the activation of WNT/*β*-catenin signaling and subsequently improves physical performance.

One limitation of our study is that we only used male flies in our experiments. We did this for practical reasons given the size of the experiment, it is clear from many studies, including those using the DGRP lines used in this study, that the phenotypic responses of females of a given genotype often do not match those of males of the same genotype [56,57]. These genotype by sex interactions can complicate our understanding of the genetic basis of phenotypic variation, as polymorphisms that contribute to male variation may differ from those affecting the same phenotype in females. As such, we must confine our conclusions only to males and future studies will be necessary to determine whether our findings can be generalized to both sexes.

In summary, our results confirm the influence of genes in the WNT signaling pathway on physical performance and support the hypothesis that expression of these genes in fly skeletal muscle affects male climbing speed, and sensitivity to drug treatment. Our use of the sensitivity index identified polymorphisms that predict the magnitude and direction of response of physical performance traits to drug treatment. The use of this index is broadly applicable for any medication and model system of interest and provides an innovative approach to advance the field of personalized medicine.

## Figures and Tables

**Figure 1 genes-13-00143-f001:**
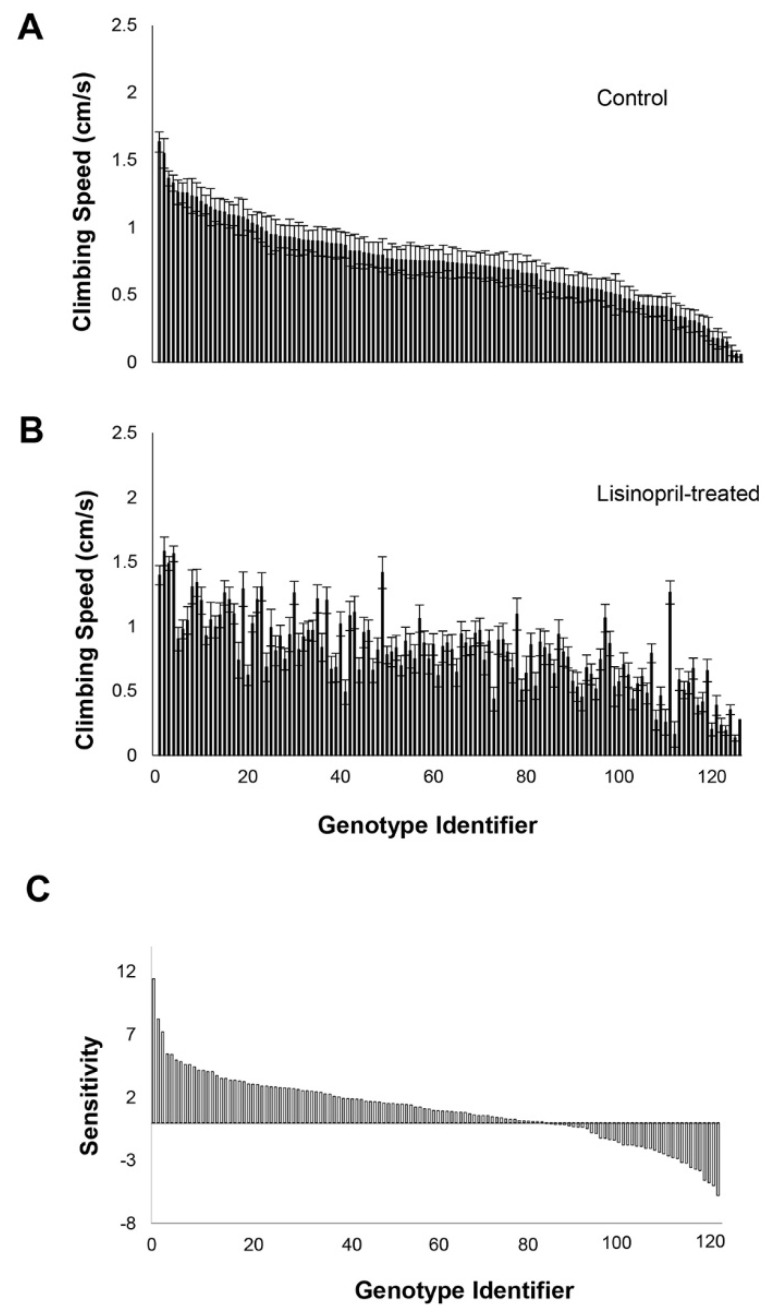
Effects of Lisinopril on climbing speed, sensitivity, and magnitude of response at old age depend on genotype. (**A**) Climbing speed at old age in flies fed control food, ranked by highest to lowest mean climbing speed. (**B**) Climbing speed at old age in flies fed Lisinopril-treated food, ranked by control at old age. (**C**) Sensitivity of climbing speed of each genotype to Lisinopril treatment varies highly at old age. “Genotype identifier” is a numbering system representing the 126 DGRP lines. These are ranked by their sensitivity, at age old age, from most to least sensitive. Positive values indicate greater speed in Lisinopril treatment relative to untreated controls; negative values indicate slower speeds relative to untreated controls.

**Figure 2 genes-13-00143-f002:**
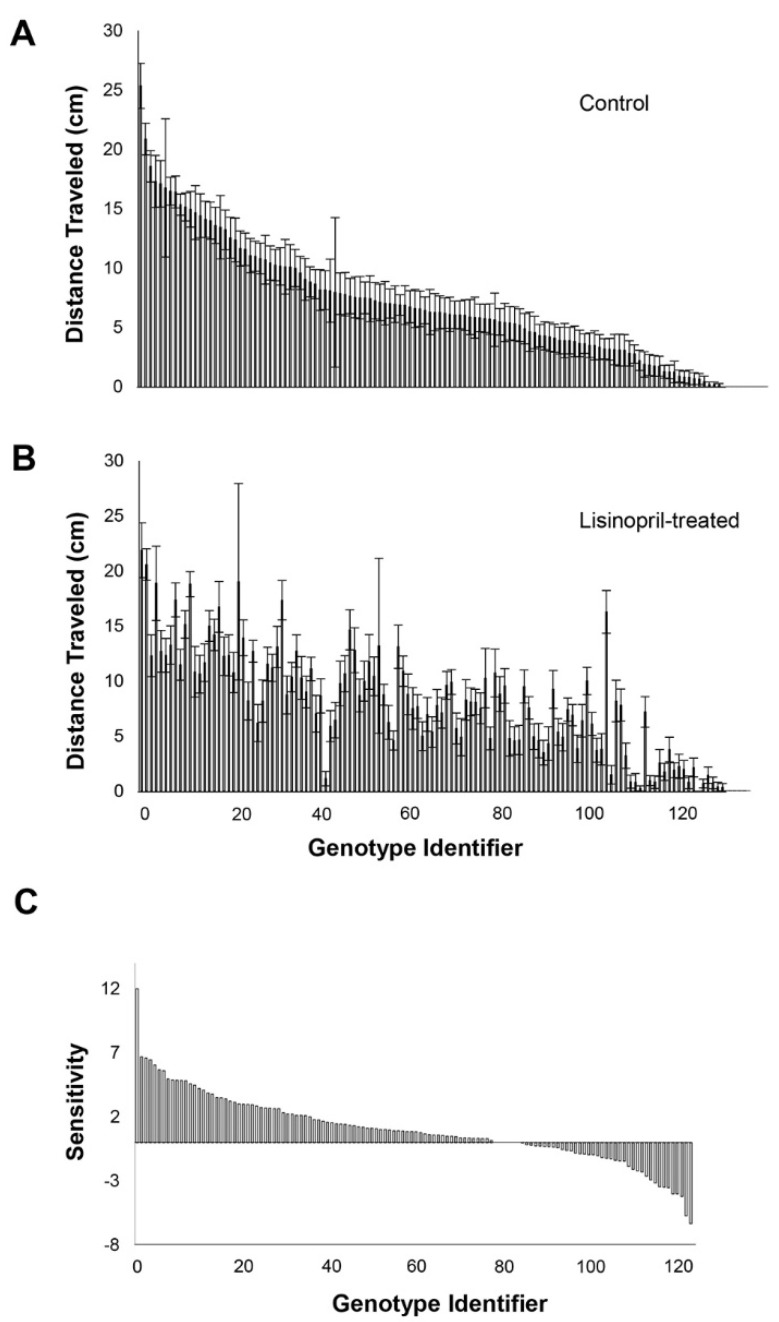
Effects of Lisinopril on endurance, sensitivity, and magnitude of response at old age depend on genotype. (**A**) Endurance (distance travelled) at old age in flies fed control food, ranked by highest to lowest mean climbing speed. (**B**) Endurance at old age in flies fed Lisinopril-treated food, ranked by control at old age. (**C**) There is significant variation among genotypes in the sensitivity of endurance to Lisinopril treatment at old age. “Genotype identifier” is a numbering system representing the 126 DGRP lines. These are ranked by their sensitivity, at age old age, from most to least sensitive. Positive values indicate greater endurance (longer distance travelled) in Lisinopril treatment relative untreated controls; negative values indicate shorter distance travelled relative to untreated controls.

**Figure 3 genes-13-00143-f003:**
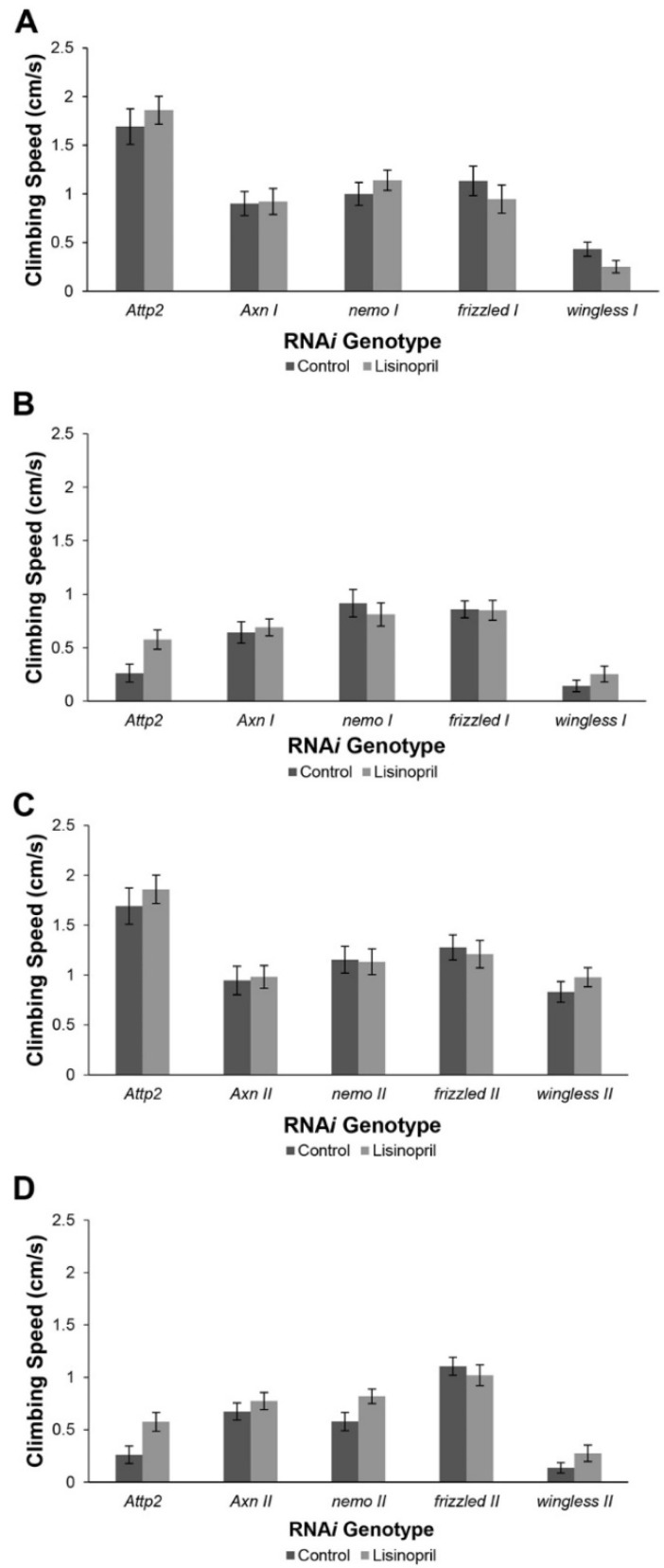
The positive effect of Lisinopril treatment on climbing speed is generally abolished in RNA*i* genotypes. (**A**) Climbing speed of control and Lisinopril-treated stock 1 of RNA*i* genotypes at 1 week of age. (**B**) Climbing speed of control and Lisinopril-treated stock 1 of RNA*i* genotypes at five weeks of age. (**C**) Climbing speed of control and Lisinopril-treated stock 2 RNA*i* genotypes at 1 week of age. (**D**) Climbing speed of control and Lisinopril-treated stock 2 RNA*i* genotypes at 5 weeks of age.

**Table 1 genes-13-00143-t001:** List of RNAi TRiP lines used to validate candidate genes and primers used to amplify cDNA for qPCR.

Gene Name and Stock	Stock Number	FlyBase Genotype	Human Ortholog
*Axin* stock 1 *Axin* primers	31705	y1 v1; P(TRiP.HM04012)attP2 F:AATGAGTGTAGTGGCCCACG R:TCTGCTACCCCTTCGGTCAT	AXIN1
*Axin* stock 2	62434	y1 v1; P(TRiP.HMJ23888)attP40/CyO	AXIN1
*frizzled* stock 1 *frizzled* primers	31036	y1 v1; P(TRiP.JF01481)attP2 F:TCTGGGACCGAACTAGATGGA R:CACGACCGGAGCAAACTGAT	FZD1
*frizzled* stock 2	34321	y1 sc * v1; P(TRiP.HMS01308)attP2	FZD1
*nemo* stock 1 *nemo* primers	41586	y1 v1; P(TRiP.GL00703)attP2 F:CTCCCTACTATCAACCGC R:GCTCCATAGCCGATAGGACGA	NLK
*nemo* stock 2	25793	y1 v1; P(TRiP.JF01799)attP2	NLK
*wingless* stock 1	31310	y1 v1; P(TRiP.JF01257)attP2 F:CCAAGTCGAGGGCAAACAGAA R:TGGATCGCTGGGTCCATGTA	WNT1
*wingless* stock 2	31249	y1 v1; P(TRiP.JF01480)attP2	WNT1

**Table 2 genes-13-00143-t002:** GWA of effect of Lisinopril treatment on physical performance and the sensitivity of age-specific climbing speed and endurance to Lisinopril treatment.

GWA Phenotype	Age (Weeks)	Treatment	Number of Indels	Number of SNPs	Number of Genes
Speed	1	Control	9	76	47
Speed	1	Lisinopril	8	127	67
Speed	5	Control	11	201	99
Speed	5	Lisinopril	1	51	29
Endurance	1	Control	2	27	9
Endurance	1	Lisinopril	2	26	9
Endurance	5	Control	23	215	72
Endurance	5	Lisinopril	5	39	14
Sensitivity of Climbing Speed	1	Lisinopril	2	23	16
Sensitivity of Climbing Speed	5	Lisinopril	6	68	27
Sensitivity of Endurance	1	Lisinopril	5	32	24
Sensitivity of Endurance	5	Lisinopril	1	40	26

## Data Availability

All data associated with this study are available in the main text or the Appendix A. All data and codes used in the statistical analyses will be deposited in a public repository for purposes of reproducing or extending the analyses upon publication.

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
