# Peer review of "Genome-Wide Analysis in Drosophila Reveals the Genetic Basis of Variation in Age-Specific Physical Performance and Response to ACE Inhibition"

_genes, 2022, doi:10.3390/genes13010143_

Round 1
Reviewer 1 Report
Here the authors examine the effect of Lisinopril, a drug used to treat high blood pressure that potentially improves physical performance in older people. They feed the drug to flies, which have the homologous target of Lisinopril. They studied the effects of the drug in 1-week old and 5-week old flies of the Drosophila Genetic Reference Panel. They examined two activity-related phenotypes: climbing ability and endurance. They find that different genotypes respond differently to Lisinopril; some genotypes exhibit an increase in activity when fed the drug, while some show a decrease in activity. They used a sensitivity measure to show that the effect of the drug on some genotypes is contrary to its intended use, i.e., some flies become less active after ingesting the drug. They conduct a GWAS to identify candidate genes responsible for the effects of Lisinopril on climbing activity and endurance. Using RNAi knockdown of candidate genes in muscles, they show that mutations in Wnt signaling genes-- Axn, nemo, frizzled, and wingless --obliterate the effects of Lisinopril. The authors also conduct GWA on the sensitivity to Lisinopril, finding that genes associated with drug treatment were different than those associated with variation in climbing ability and endurance. This is a well-motivated and executed study that identifies important considerations for personalized medicine, inspiring future work in this area. I have the following comments.
The editors should note that for some reason, when I downloaded the supplementary zip file, it was empty. I was not, therefore, able to evaluate the supplementary materials.
Comments
- General: Restricting the behavioral tests and GWA to male flies is a limitation of this study and should be discussed briefly.
- Methods, Lines 138-142: The GWA on sensitivity measures used a two-factor model. It would be helpful to state the model explicitly as the DGRP2 webtool was not used in this case. Does the sensitivity analysis take chromosomal inversions and Wolbachia infection into account as the DGRP2 webtool does? What software package was used for these calculations?
- Methods, Lines 146, the authors state that the GWA tested 4.8 million SNPs, but prior reports using the DGRP2 webtool suggest that the actual number of SNPs tested is considerably less than this number as some SNPs are unique to each genotype. How many SNPs were tested?
- Methods, Lines 91-94, were the flies fed Lisinopril from egg to the 1-week and 5-week assays, so that the dosage was chronic?
- Discussion, Lines 464-477, the authors state that “genes in the WNT signaling pathway are strongly implicated as contributors to variation in climbing speed and the sensitivity to drug treatment.” This assertion is borne out by the network analyses, the candidate gene tests and their previous expression data. However, the attP2 control had a significant response to Lisinopril for one experiment (Fig. 3B) but not the other (Fig. 3D). The genetic background may be part of the reason the RNAi knockdown lines are non-responsive, a limitation that should be discussed.
Author Response
Reviewer 1 – General comments
“Here the authors examine the effect of Lisinopril, a drug used to treat high blood pressure that potentially improves physical performance in older people. They feed the drug to flies, which have the homologous target of Lisinopril. They studied the effects of the drug in 1-week old and 5-week old flies of the Drosophila Genetic Reference Panel. They examined two activity-related phenotypes: climbing ability and endurance. They find that different genotypes respond differently to Lisinopril; some genotypes exhibit an increase in activity when fed the drug, while some show a decrease in activity. They used a sensitivity measure to show that the effect of the drug on some genotypes is contrary to its intended use, i.e., some flies become less active after ingesting the drug. They conduct a GWAS to identify candidate genes responsible for the effects of Lisinopril on climbing activity and endurance. Using RNAi knockdown of candidate genes in muscles, they show that mutations in Wnt signaling genes-- Axn, nemo, frizzled, and wingless --obliterate the effects of Lisinopril. The authors also conduct GWA on the sensitivity to Lisinopril, finding that genes associated with drug treatment were different than those associated with variation in climbing ability and endurance. This is a well-motivated and executed study that identifies important considerations for personalized medicine, inspiring future work in this area. I have the following comments.”
Response: We thank the reviewer for the kind comments on our work.
“The editors should note that for some reason, when I downloaded the supplementary zip file, it was empty. I was not, therefore, able to evaluate the supplementary materials.”
Response: We apologize for this issue. Unknown to us when we submitted this material, there was a formatting issue with the zip file. We have corrected this issue and the current version of the manuscript now has all of the supporting documentation correctly formatted and available for review.
Reviewer 1 - Specific Comments
- “General: Restricting the behavioral tests and GWA to male flies is a limitation of this study and should be discussed briefly.”
Response: We agree and thank the reviewer for pointing this out. We note that our decision to use only males was a practical one, and based purely on the size of experiment. To address the concern of the reviewer we have revised the manuscript to acknowledge the ubiquity of genotype by sex interactions and acknowledge this limitation of our study on lines 358-365.
- “Methods, Lines 138-142: The GWA on sensitivity measures used a two-factor model. It would be helpful to state the model explicitly as the DGRP2 webtool was not used in this case. Does the sensitivity analysis take chromosomal inversions and Wolbachia infection into account as the DGRP2 webtool does? What software package was used for these calculations?”
Response: We apologize for the confusion here. To clarify, the GWA on sensitivity is exactly the same as GWAS for any phenotype except that the phenotype is now a sensitivity index and we used the DGRP2 web tool for the analysis. Because the DGRP2 web tool allows researchers to input data on two phenotypes, (e.g, one phenotype column for each sex or perhaps two environments) we took advantage of the structure of this analysis to use the trait phenotype as one factor and the sensitivity value of that line as the other. We have rewritten this section of the methods to more clearly explain exactly what we did in these analyses (lines 429-435). Because these GWA analyses used the DGRP2 web tool, inversions and Wolbachia infection status were accounted for.
- “Methods, Lines 146, the authors state that the GWA tested 4.8 million SNPs, but prior reports using the DGRP2 webtool suggest that the actual number of SNPs tested is considerably less than this number as some SNPs are unique to each genotype. How many SNPs were tested?”
Response: We thank the reviewer for pointing out this mistake. The 4.8 were total SNPs available in the DGRP and indeed we only tested 1887374 SNPs in this study. We have corrected this (line 439).
- “Methods, Lines 91-94, were the flies fed Lisinopril from egg to the 1-week and 5-week assays, so that the dosage was chronic?”
Response: Flies treated with Lisinopril were maintained on the Lisinopril over the entire course of each experiment. We have added this detail (line 378-379).
- “Discussion, Lines 464-477, the authors state that “genes in the WNT signaling pathway are strongly implicated as contributors to variation in climbing speed and the sensitivity to drug treatment.” This assertion is borne out by the network analyses, the candidate gene tests and their previous expression data. However, the attP2 control had a significant response to Lisinopril for one experiment (Fig. 3B) but not the other (Fig. 3D). The genetic background may be part of the reason the RNAi knockdown lines are non-responsive, a limitation that should be discussed.”
Response: The Harvard TRiP lines that we used were designed so that the knockdown lines share the same genetic background as that of the control and other RNAi lines (https://fgr.hms.harvard.edu/trip-rnai-control-fly-stocks), with the exception that the control line is missing the RNAi construct. The lack of a significant effect on the control line at younger age we suspect is a power issue. However, we note that in every case, the control lines climbed faster on Lisinopril than those fed control food. We also agree however that the reviewer is right to point out that there is some inconsistency between the responses of the RNAi lines. We have revised the manuscript to more fully explain our results (lines 255-263, lines 338-339).
Reviewer 2 Report
The article entitled “Genome-wide analysis in Drosophila reveals the genetic basis of variation in age-specific physical performance and response to ACE inhibition” the authors present compelling evidence that genes of the WNT signalling pathway modulate physical (muscle) performance. The article is well written and is easy to understand. Nevertheless, I would like to see some more details regarding the RT-PCR experiment, namely:
- Age of flies used in the RT-PCR experiment
- What are the primer efficiencies?
- How was RNA quality evaluated?
- What is the percentage of inactivation that is achieved using the driver lines?
This last point is important because the authors state that “The endurance of all 16 untreated RNAi-gene lines closely resembles that of Lisinopril-treated flies for all six RNAi stocks at old age (eFigure 9)” which could imply that the driver used is a weak driver that only suppresses slightly the expression of the target genes.
An explanation should also be offered for why only virgin males were used (line 90).
Some other minor corrections:
- Line 57-58. Check the grammar.
- Line 59-60. Make clear whether you are referring to the Drosophila or Human gene.
- Line 190. Please state that old age means 5 weeks.
- Line 235 and 244. “at age old age”. Please clarify.
- Line 246. Check the grammar.
- Line 383. Check the grammar.
- Line 436. Check the grammar.
Tables go across more than one page. Why are Figure and Tables called eFigures and eTables when referred in the text, if they are named Figure and Tables?
Author Response
Reviewer 2
Comments and Suggestions for Authors
“The article entitled “Genome-wide analysis in Drosophila reveals the genetic basis of variation in age-specific physical performance and response to ACE inhibition” the authors present compelling evidence that genes of the WNT signaling pathway modulate physical (muscle) performance. The article is well written and is easy to understand.”
Response: We thank the reviewer for the kind comments.
“Nevertheless, I would like to see some more details regarding the RT-PCR experiment, namely:
- Age of flies used in the RT-PCR experiment”
Response: All flies used in the RT-PCR experiment were one week of age. We have added this information to the manuscript (line 496).
- “What are the primer efficiencies?”
Response: Unfortunately we do not know the primer efficiencies. This is something we don’t typically do unless we are investigating different primer sets or are having difficulties with the method. We appreciate the concern, but given that measured expression was high in some conditions and we used the same primers for each gene across each line we suspect that even if the efficiencies were low, they would be low in all genotypes. As such, this should not bias our results.
“3. How was RNA quality evaluated?”
Response: We typically only do this if we suspect that the quality of the RNA has degraded over time. In this case we didn’t check the RNA quality in our experiments because we kept the samples on ice at all times during the extraction protocol, stored the RNA at -80C, and used the RNA soon after it was extracted.
- “What is the percentage of inactivation that is achieved using the driver lines? This last point is important because the authors state that “The endurance of all 16 untreated RNAi-gene lines closely resembles that of Lisinopril-treated flies for all six RNAi stocks at old age (eFigure 9)” which could imply that the driver used is a weak driver that only suppresses slightly the expression of the target genes.”
Response: This is a good point – while this information was in the supplemental files there was a problem in the formatting of those files and so they were not accessible to the reviewers. We didn’t know this at the time and apologize for this error. We have now reformatted these files and they are available for review. The data requested in this case can be found in “Data file S7…”. The efficiency of the knockdown varied depending on the gene and this information is contained within this file for each gene. The knockdown ranged from 0.25 fold (i.e., 75% for frizzled, wingless) to 0.79 fold (21% for nemo). We now also mention this in the manuscript (line 471-472).
“An explanation should also be offered for why only virgin males were used (line 90).”
Response: We used virgin males to avoid the complicating physiological effects of reproduction and mating frequency on phenotypic traits. We have added this information to the manuscript (Lines 373 – 374).
“Some other minor corrections:
- Line 57-58. Check the grammar.”
Response: fixed
- “Line 59-60. Make clear whether you are referring to the Drosophila or Human gene.”
Response: fixed
“Line 190. Please state that old age means 5 weeks.”
Response: done. We also changed the figure legend to specify one and five weeks of age.
“Line 235 and 244. “at age old age”. Please clarify.”
Response: fixed
“Line 246. Check the grammar.”
Response: fixed
“Line 383. Check the grammar.”
Response: fixed
“Line 436. Check the grammar.”
Response: fixed
“Tables go across more than one page. Why are Figure and Tables called eFigures and eTables when referred in the text, if they are named Figure and Tables?”
Response: We apologize for the confusion. We had labeled the eTables as those that were in the supplemental files, which unfortunately the reviewers did not have access to. We have now relabeled all supplemental files as either Data file S# (these are excel spreadsheets), Table S#, or Fig. S# to more clearly indicate that these files are part of the supplemental information.
Reviewer 3 Report
In this research paper, the author’s goal is to identify genes or signaling networks implicated in physical performances improvement observed upon Lisinopril treatment, an inhibitor of the angiotensin-converting enzyme (ACE). They use Drosophila as a model system to perform a GWAS study to identify candidate genes contributing to natural variations in age-dependent physical performance in Lisinopril-treated flies. Interestingly, this approach led to the identification to many genes of the WNT signaling pathway.
The study is well designed and is overall convincing. The data are nicely explained and will be of potential interest for the field. In addition, this work can be easily used as a resource article. The text is well written and is easily understandable even for non-specialists of this type of genetic approach.
Even though I would be very positive for the acceptance of the paper, but in my opinion a crucial experiment is missing in the manuscript in its current version. The author mentioned that previous structural studies showed that Lisinopril binds to Drosophila ACE (ance). They partially documented ance loss of function adult muscle in a previous paper by evaluating the impact on lifespan on Lisinopril-treated flies, but no assessment of their physical performances. Therefore, it would be essential to document the impact of Lisinopril treatment on climbing speed and distance travelled in the context of muscle specific-invalidation of ance. this would represent an important validation of the model. This is particularly important in the context of their proposed results in which differences between control and Lisinopril-treated stocks of RNAi genotypes are limited and at the limit of statistical significance.
Author Response
Reviewer 3
Comments and Suggestions for Authors
“The study is well designed and is overall convincing. The data are nicely explained and will be of potential interest for the field. In addition, this work can be easily used as a resource article. The text is well written and is easily understandable even for non-specialists of this type of genetic approach.”
Response: We thank the reviewer for the kind comments.
“Even though I would be very positive for the acceptance of the paper, but in my opinion a crucial experiment is missing in the manuscript in its current version. The author mentioned that previous structural studies showed that Lisinopril binds to Drosophila ACE (ance). They partially documented ance loss of function adult muscle in a previous paper by evaluating the impact on lifespan on Lisinopril-treated flies, but no assessment of their physical performances. Therefore, it would be essential to document the impact of Lisinopril treatment on climbing speed and distance travelled in the context of muscle specific-invalidation of ance.”
Response: We appreciate the point raised by the reviewer and indeed this set of experiments would have been useful to have done. However, we feel that based on the prior knowledge of the binding affinity of Lisinopril to ance in flies, our demonstration that ance knockdown and Lisinopril treatment mimics each others’ effects on life span, and our demonstration of genetically based differences in response in physical performance traits (speed, endurance and strength) to the drug is sufficient justification for our study. We do acknowledge however that the effects of Lisinopril on physical performance could be indirect, and indeed this could be the case in humans as well. We feel that as it stands our study outlines a general approach for understanding the broader question of what is the genetic basis of variation in drug response, and identifies an important pathway that could be involved in the response to this particular drug. We hope that this satisfies the concern of this reviewer.
Round 2
Reviewer 3 Report
Even though they do not address specifically my concern, overall I agree with the authors' comments and I think that this manuscript can be accepted in its present form.